# Impact of Health Access on Hospitalization of Children and Youth with Cerebral Palsy and Paralytic Syndromes in Brazilian Regions

**DOI:** 10.3390/ijerph21101286

**Published:** 2024-09-26

**Authors:** Silvia Baltieri, Rubens Wajnsztejn

**Affiliations:** 1ABC Medical School, Santo André CEP 09060-650, SP, Brazil; 2Scientific Writing Laboratory, Specialized Learning Center, Graduate Program in Health Sciences, ABC Medical School, Santo André CEP 09060-650, SP, Brazil; rubens.wajnsztejn@fmabc.br

**Keywords:** Unified Health System, socioeconomic factors, cerebral palsy

## Abstract

In 2021, 28.16% of Brazilians were aged 0–19 years, highlighting the critical need for medical care in diagnosing and treating cerebral palsy (CP). In developing countries, CP prevalence reaches 7/1000 live births, emphasizing the importance of healthcare access, which influences diagnosis and prognosis. This study aimed to analyze the correlation between health infrastructure across different Brazilian regions and its impact on hospitalization and mortality rates among children with CP and other paralytic syndromes. An ecological time-series design was used, analyzing secondary population-based data from DATASUS, covering January 2018 to December 2021. The data included healthcare facilities, physicians, hospitalizations, and deaths for individuals aged 0–19 years, with analysis conducted using Microsoft Excel. The results revealed significant disparities in healthcare infrastructure, particularly in the northern and northeastern regions. A notable correlation was found between healthcare inequalities and hospitalizations and mortality rates, with the northern region showing significant results (*p* = 0.03 for hospitalizations and *p* = 0.02 for mortalities). This study underscores significant regional disparities in healthcare access in Brazil, contributing to variations in hospitalization rates for children and adolescents, especially in the northern region.

## 1. Introduction

Brazil’s geographic disparities result in sociospatial and socioeconomic inequalities, along with an uneven distribution of healthcare resources. The COVID-19 (Coronavirus Disease 2019) pandemic has exacerbated these disparities, leading to a systemic and prolonged economic and social crisis in areas with greater sociospatial inequality [1]. The age range of 0–19 years was selected for this study because pediatric care is a priority due to children’s susceptibility to illness and the potential for diseases to worsen during this vulnerable life stage [2].

The period from 2018 to 2021 included two years prior to the COVID-19 pandemic and the first two years of the pandemic, a time when the Unified Health System (SUS) was a focal point of public debate alongside the monitoring of cases and death rates. Comparing healthcare service utilization data before and after the onset of the COVID-19 pandemic is crucial for understanding shifts in healthcare demands and service delivery patterns [3]. The official records from DATASUS, as presented in this study, highlight how the variables affected access to healthcare infrastructure for children and adolescents across different Brazilian regions.

Respiratory diseases are one of the main complications associated with the COVID-19 virus. In the DATASUS system, cerebral palsy (CP) is a measurable condition in the specified age group that frequently presents with respiratory impairment. Thus, we chose to investigate healthcare access for all individuals within this age range (considered the control group) and for those with CP (considered the specific group).

CP is a neurodevelopmental disorder characterized by changes in muscle tone, movement, and motor skills, attributed to nonprogressive lesions in the developing brain [4,5]. Some impairments, such as CP, become apparent only as the child grows and develops, making diagnosis and epidemiological documentation more challenging [6].

In developing countries, the prevalence of CP reaches 7 per 1000 live births when considering all levels of severity [7,8,9]. In Europe, a report that used standardized definitions and included 6000 children with CP from 13 geographically defined populations found an overall rate of 2.08 per 1000 live births [10]. According to the Brazilian Cerebral Palsy Association [11], 17 million people worldwide live with this condition.

In Brazil, there is a scarcity of research specifically examining the prevalence and incidence of CP at a national level. However, based on data from other countries, estimates of CP prevalence in developing countries can be projected [12]. The Census does not include specific questions about this diagnosis but only about physical disabilities, which have different prognoses and treatments.

The disparity in prevalence between these two groups of countries is attributed to inadequate prenatal care and primary healthcare for pregnant women [10]. This issue likely extends to postnatal medical follow-ups, hindering the observation of early childhood developmental milestones and, consequently, opportunities for early intervention. Therefore, a child with a suspected diagnosis of CP requires constant medical supervision, particularly during early childhood and especially in Primary Health Care Units. Additionally, specific analyses are essential for a medical team to confirm the diagnosis and register it in the DATASUS.

This study enabled the correlation of data to elucidate the current healthcare access situation for individuals diagnosed with CP. CP is a chronic condition characterized by a set of nonprogressive impairments affecting various systems, notably the respiratory system. Common respiratory issues include upper and lower airway (UA-LA) pathologies, oropharyngeal incoordination (OIC), secretion accumulation, bronchitis, recurrent pneumonia, and gastroesophageal reflux (GER). GER is particularly disabling as it frequently leads to aspiration pneumonia. Children with CP are highly susceptible to opportunistic respiratory diseases []. While the primary symptoms of COVID-19 are respiratory, its clinical presentation can diverge significantly. During the pandemic, individuals with disabilities were at a higher risk of developing comorbidities [13].

This study aimed to examine the relationship between structural health inequalities and their impact on hospitalization and mortality rates among children and adolescents with cerebral palsy and other paralytic conditions across different regions of Brazil.

## 2. Method

### 2.1. Study Design

This population-based ecological time-series study used secondary data from DATASUS, the Brazilian Ministry of Health database.

### 2.2. Study Location and Timeframe

According to the Brazilian Institute of Geography and Statistics (IBGE) [14], Brazil’s land area in 2021 was 8,510,345.540 km^2^, with a population of 214.3 million, of which 60,359,871 were aged 0–19 [15]. The country had a population density of 25.07 inhabitants/km^2^, a Human Development Index (HDI) of 0.754, and a Gini coefficient of 0.544. To assess healthcare disparities across this vast territory, this study was conducted regionally, focusing on the 0–19 age group.

In 2021, the northern region had 6,675,464 [15] children and young people in the specified age group, a population density of 4.1 inhabitants/km^2^ [14], an HDI of 0.667 [15], and a Gini coefficient of 0.528 [15]. The northeast region had 18,279,330 [15] people in the specified age group, a population density of 34.1 inhabitants/km^2^ [14], an HDI of 0.663 [15], and a Gini coefficient of 0.556 [15]. The midwest region had 4,761,462 [15] people in the specified age group, a population density of 10.01 inhabitants/km^2^ [14], an HDI of 0.757 [15], and a Gini coefficient of 0.514 [15]. The southeastern region had 22,880,390 [15] people in the specified age group, a population density of 84.21 inhabitants/km^2^ [14], an HDI of 0.766 [15], and a Gini coefficient of 0.533 [15]. The southern region had 7,763,225 [15] people in the specified age group, a population density of 53.19 inhabitants/km^2^ [14], an HDI of 0.754 [15], and a Gini coefficient of 0.462 [15].

### 2.3. Study Population and Eligibility Criteria

The population consisted of individuals aged 0–19 years, using data from TABNET, which were tabulated in Microsoft Excel (Microsoft 365). The data include the number of healthcare facilities, practicing physicians, hospitalizations, and mortalities of children and young people with cerebral palsy and other paralytic syndromes across all diagnostic categories. This study covered the period from 2018 to 2021, which included two years prior to the COVID-19 pandemic and the first two years of the pandemic in Brazil. Sociodemographic variables considered were sex (male, female, and unspecified); age (under 1 year, 1–4 years, 5–9 years, 10–14 years, 15–19 years); race/ethnicity (white, black, mixed race, Asian, indigenous, not reported); region by place of residence, hospitalization, and death occurrence (north, south, midwest, northeast, and southeast); and healthcare system (public, private, and unknown).

### 2.4. Data Collection

#### Data Stratification

Secondary data were collected from the Brazilian National Registry of Health Establishments (CNES), Human Resources—Professionals—Individuals (CBO 2002), the Hospital Information System of the Unified Health System (SIH/SUS), and the Mortality Information System (SIM) from the Department of Informatics of the Unified Health System (DATASUS) website. Population projections for Brazil and its Federative Units from 2000 to 2030, provided by the Brazilian Institute of Geography and Statistics (IBGE), were also used. All data were accessed through https://datasus.saude.gov.br/, accessed on 10 December 2022.

The strategy used to access, tabulate, and retrieve the data involved the following steps. For demographic data, we accessed https://datasus.saude.gov.br > Information Access > TABNET Health Information > Demographics and Socioeconomics > Resident Population > State Population Projection by Sex and Age Groups: 2000–2030. Within the tab interface, we selected the following options: Column: years 2018 to 2021; Row: Age group 2 from 0 to 19 years; Content: population. Variable selection > Sex: Male or female. Data from each region (north, northeast, southeast, south, and midwest) were collected and analyzed separately, along with a comprehensive analysis across all regions.

To collect data on healthcare facilities, we selected CNES—Facilities > Facility Types > Geographic Scope > Brazil by Region, State, and Municipality under the health system option. We then filtered the following options: 2018 to 2021; content: number of health facilities in each Brazilian region; row: north, northeast, southeast, south, and midwest regions.

For data on physicians, we selected human resources as of August 2007: professionals classified by CBO 2002 > Professionals > Geographic Coverage > Brazil by Region, State, and Municipality under the health system option. We filtered the following options: years 2018 to 2021; row: north region, northeast region, southeast region, south region, and midwest region; content: number of physicians in each Brazilian region.

To collect data on hospitalizations by place of residence, we selected Epidemiology and Morbidity > SUS Hospital Morbidity (SIH/SUS) > General by place of residence—from 2008 > Brazil by Region and State. We filtered the following options: years 2018–2021; row: north region, northeast region, southeast region, south region, and midwest region; content: hospitalizations considering variables such as age 0–19 years; type of care > all categories; healthcare system > public, private, not specified; sex > male, female, unknown > race/ethnicity > white, black, mixed race, Asian, indigenous, and unreported. Hospitalization was assessed using two independent surveys: ICD-10 Chapter All categories ICD-10 Morbidity List All types and ICD-10 Morbidity List Cerebral palsy and other paralytic syndromes. The same procedure was followed to collect data on hospitalization by place of admission.

For data on deaths by place of residence, we selected Vital Statistics > Mortality since 1996 according to ICD-10 > General mortality > in Brazil by Region and State. We filtered the following options: years 2018 to 2020; row: north region, northeast region, southeast region, south region, and midwest region; content: deaths considering the following variables such as age 0–19 years; sex > male, female, unknown > race/ethnicity > white, black, brown, Asian, indigenous, no information > place of occurrence > hospital, other health facilities, home, public road, and other, unknown. Deaths were assessed using two independent surveys: ICD-10 Chapter > All categories > Causes—ICD-BR-10 > All categories and ICD-10 Group > Cerebral palsy and other syndromes > ICD-10 Category > G80 Cerebral palsy and G83 other paralytic syndromes. The same procedures were followed to collect data on deaths by place of occurrence.

### 2.5. Data Analysis

Descriptive analysis was conducted to report and characterize the events and variables of interest. The rate of healthcare facilities and health professionals by region was calculated annually from 2018 to 2021 as follows: the number of facilities in each region was divided by the population aged 0–19 years in that region and then multiplied by 100,000. The same calculation was applied to measure the number of physicians per region and year. This approach preserved the respective population proportions, thereby minimizing the impact of external variations throughout the evaluated period.

To calculate hospitalization rates by region, the number of hospitalizations for individuals aged 0–19 years, across all disease categories, was divided by the total population of that age group and multiplied by 100,000. The same calculation was used to measure hospitalizations for individuals in the same age group with a specific diagnosis of CP or other paralytic syndromes (Appendix A).

To calculate mortality rates by region, deaths among individuals aged 0–19 years across all disease categories were tallied, divided by the total population of that age group, and multiplied by 100,000. The same calculation was used to measure deaths among individuals of the same age group with a specific diagnosis of CP or other paralytic syndromes (Appendix A).

For statistical analysis, the Prais–Winsten generalized linear regression model was employed, with years as the independent variable and service characteristics as the dependent variables. The Pearson’s correlation test was then applied. The confidence level was set at 95%, and statistical analyses were performed using Stata version 16.0^®^ (StataCorp, College Station, TX, USA).

The statistical analysis demonstrating trends in healthcare access during the study period is presented in Appendix A (attached).

Appendix A illustrates the evolution of health indicators in Brazil between 2018 and 2021. It presents the annual percentage change (APC) in key metrics, including the number of establishments, healthcare professionals, hospitalizations, and deaths, disaggregated by region. A further analysis is presented according to category:Healthcare establishments:

All regions exhibited an upward trajectory in the number of healthcare establishments, with a notable positive acceleration (*p* < 0.001 for the majority of regions).

The south exhibited the highest APV (935.36), while the north demonstrated the lowest growth (261.77), both with an upward trend.

The Durbin–Watson test indicates that the values are within an acceptable range, suggesting that there is no significant autocorrelation in the model residuals, with the exception of the southeast, which exhibited a value of 2.95, indicating the potential for serial correlation.

2.Number of doctors:

Furthermore, all regions demonstrated a notable increase in the number of medical practitioners, with an APV ranging from 297.4 (in the north) to 955.43 (in the southeast).

As with establishments, the southeast and south have the highest growth rates in the number of doctors.

All Durbin–Watson values indicate an upward trend.

3.Hospitalizations for all conditions (Hosp_ALL_place):

All regions exhibited a notable increase in hospitalizations for all pathologies, with the north region demonstrating the most pronounced upward trend, as indicated by the highest APV (3951.83), and the south exhibiting the least pronounced trend (3990.69).

The Durbin–Watson value for all regions indicates the absence of serial correlation, with a consistent upward trend.

4.Hospitalizations for cerebral palsy (Hosp_CP_place):

The data indicate an upward trend in hospitalizations for cerebral palsy in nearly all regions, with the exception of the northeast, where a significant downward trend was observed (APV = 4.7; *p* < 0.001).

The midwest region exhibited the most pronounced percentage variation (APV = 10). The Durbin–Watson test indicates the potential for serial correlation in the northeast region (value of 2.66), which merits further investigation.

5.Mortality from all causes (Dea_ALL_place):

Furthermore, a similar upward trend was observed in deaths from all pathologies in the north, northeast, and central-west regions, with the highest APV in the north (97.95).

In the southeast and south, the trend is stable, with Durbin–Watson values also indicating the absence of autocorrelation in these cases.

It is noteworthy that, although the APV for these regions is lower, the values are nevertheless marginally significant, suggesting a stabilization in deaths.

The data indicate a trend towards an increase in both health infrastructure and the number of doctors in all regions of Brazil, which reflects ongoing investments in the health sector. However, the increases in hospitalizations and deaths, especially in the north and northeast regions, suggest that these structural advances have not yet been sufficient to reverse health conditions in these areas, particularly in the context of chronic diseases such as cerebral palsy.

The stabilization of mortality rates in the southeast and south may indicate a reduction in risk factor control or an improvement in the management of chronic diseases such as cerebral palsy. However, the continued growth in hospitalizations for cerebral palsy in the midwest requires further investigation, as it may reflect improvements in diagnosis or an increase in incidence.

### 2.6. Ethical and Legal Aspects of Research

As this study used data from publicly accessible secondary databases, submission to or approval by a Research Ethics Committee was not required in accordance with the National Health Council’s regulations for research involving human subjects (Resolution CNS 466/12).

## 3. Results

The official population projections for individuals aged 0–19 from 2018 to 2021, as released by the IBGE and available on DATASUS (see Appendix A), alongside the percentage of healthcare facilities and physicians in each region relative to the national total, reveal significant disparities. When comparing the proportion of the population to the availability of healthcare facilities and professionals, the north and northeast regions, despite having larger populations, possess significantly fewer health resources than other regions.

The percentage calculations considered the number of healthcare facilities and physicians per 100,000 inhabitants in the 0–19 age group, maintaining the respective population ratios (see Appendix A). Although there was a temporal trend of an increase in healthcare facilities and physicians—ranging from 4% to 11% across all regions—this increase did not sufficiently address the disparities in areas with fewer facilities.

To analyze regional disparities, we focused on comparing the north and northeast regions with other areas. Appendix A reveals significant differences in healthcare infrastructure. During the analyzed period, it was observed that the north had 3% more inhabitants than the midwest, yet the latter had 103% more healthcare facilities and 138% more doctors. The southeast had 27% more inhabitants than the north but 163% more healthcare facilities and 228% more doctors. Similarly, the south had 2% more residents than the north but 241% more healthcare facilities and 196% more doctors. When comparing the southeast to the northeast, the former had 8% more people but 87% more healthcare facilities and 133% more doctors. Finally, although the northeast’s population was 17% larger than the south’s, the south had 158% more healthcare facilities and 112% more doctors.

These disparities are confirmed by the statistical analyses presented in Appendix A (attached).

The regional disparities in Brazilian healthcare infrastructure are shown in Appendix A. By correlating the Human Development Index (HDI), obtained from the Brazilian Human Development Atlas [16], with COVID-19 vaccination data provided by Fiocruz and DATASUS, it becomes evident that regions with lower HDI scores have less developed healthcare infrastructures [16]. The disparity between Brazilian regions is evident in Appendix A (both attached).

The growth of inequalities is a characteristic phenomenon of the globalization era, manifesting in varying degrees across countries and societies [17]. Brazil is the seventh most unequal country globally [18], with inequality intensifying in recent years [19]. Studies analyzing wealth data conclude that income concentration among Brazil’s wealthiest segment has not declined, and indeed, it has increased [20,21]. The richest 1% holds 48% of all national wealth, whereas 50% of the population owns only approximately 3% of the country’s total wealth.

The Human Development Index (HDI) is a composite measure of development based on life expectancy, education, and per capita income. Countries with higher HDI scores tend to have better healthcare systems, superior education, and a higher overall quality of life. Conversely, lower HDI scores often correlate with challenges in providing adequate healthcare, potentially limiting access to vaccinations, hospital care, and pediatric treatments [22]. The Gini Index, which measures income inequality within a population, further illustrates this disparity. It ranges from zero to one, with zero representing perfect equality (everyone has the same income). A value of one represents the opposite extreme (maximum inequality). In the 2004 Human Development Report by the United Nations Development Program, Brazil had a Gini Index of 0.591, ranking near the bottom of 127 countries [23].

Social inequality, as reflected by the Human Development Index (HDI) and Gini Index, can significantly affect access to healthcare by influencing factors such as vaccination rates and hospital admissions, including those for children. The Gini Index^16^ in Brazil was 0.545 in 2018, 0.544 in 2019, 0.524 in 2020, and 0.544 in 2021. The HDI^16^ in Brazil in 2010 was 0.699 (Appendix A).

### 3.1. Admissions

Encompassing all disease categories, the total number of hospitalizations by place of treatment was 75,459, whereas the total number by place of residence was 75,580. In the category “Hospitalizations for Cerebral Palsy and other paralytic syndromes”, the total number of admissions by treatment location was 98, compared to 94 by patient residence. The discrepancy in the total number of hospitalizations across categories may indicate underreporting, which is a limitation of this study. For the annual analysis comparing regions, data on hospitalizations by place of care were used, as they provided a more comprehensive dataset.

Appendix A reveals a significant discrepancy in the northern region when comparing the hospitalizations of individuals diagnosed with CP and other paralytic syndromes by hospitalization and residence. The number of hospitalizations in this region was notably lower than that of residents with these conditions. In the midwest region, there was also a significant difference (27.5%) between the place of hospitalization and place of residence; however, the figures for hospitalization were higher. This analysis suggests that individuals from the north may have migrated to other regions in search of hospitals and medical care. However, this conclusion is imprecise because the discrepancy in total hospitalization numbers makes it difficult to determine exactly which regions treated patients who were transferred from other areas.

Moreover, when surveying the OAMT, a program providing healthcare services outside the patient’s region of residence, it was not possible to isolate data for the specific age groups studied in this research. The available documents only report aggregate figures for the entire population, regardless of age.

To obtain more detailed information on the hospitalization system (public, private, or unknown), a search was conducted in each region. However, only the available data were categorized as “unknown”.

The total number of hospitalizations for all conditions across the five regions was 2,563,560 in 2018, 2,567,688 in 2019, 1,903,448 in 2020, and 1,925,568 in 2021, amounting to 8,960,264. Of these hospitalizations, 29% occurred in 2018, 29% in 2019, 21% in 2020, and 21% in 2021. Thus, according to the data published in DATASUS, there was a higher number of hospitalizations in this age group during the two years preceding the pandemic.

When analyzing the percentage of hospitalizations across all disease categories, the variation between regions ranged from 1% to 2%. Adolescents aged 15–19 years accounted for 31% of the hospitalizations, followed by infants under 1 year (26%), children aged 1–4 years (19%), children aged 5–9 years (13%), and children aged 10–14 years (11%). Women accounted for the majority of hospitalizations (54%).

Regarding race/ethnicity, individuals who self-identified as mixed race accounted for 44% of hospitalizations, whereas those who did not report their race/ethnicity accounted for 26%. The lack of data on race/ethnicity in the DATASUS represents a limitation of this study.

Regarding the nature of admissions, emergency care accounted for 86%.

Comparing the percentage of hospitalizations for individuals with CP and other paralytic syndromes across Brazilian regions in 2018, significant disparities were evident. The southern region had 140% more CP-related hospitalizations than the northern region. The midwestern region exhibited an even more striking difference, with 800% more hospitalizations than the northern region. The northeast and southeast regions also reported substantially higher rates, with 380% and 440% more hospitalizations, respectively, than the north region. These figures indicate that when CP was diagnosed, the number of recorded hospitalizations in the northern region was significantly lower than that in the other regions.

Children aged 5 to 9 accounted for 30% of hospitalizations, those aged 10 to 14 for 26%, followed by adolescents aged 15 to 19 (22%), children aged 1 to 4 (19%), and infants aged < 1 year (4%). Males represented the majority of hospitalizations (57%).

In terms of race/ethnicity, 64% of the hospital admission records lacked this demographic information. The absence of race/ethnicity data in DATASUS represents a limitation of this study. Regarding the nature of hospitalizations, elective procedures comprised 85% of admissions.

### 3.2. Deaths

Data on deaths from the Mortality Information System (SIM)^18^ cover up to 2020.

As shown in Appendix A, the north and northeast regions reported the highest mortality rates across all categories on the Morbidity List. No regional discrepancies were observed in deaths among patients aged 0–19 with CP and other paralytic syndromes. For the annual regional comparison analysis, data on deaths by place of occurrence were used to maintain consistency with the hospitalization approach. The total number of deaths from all causes across the five regions was 68,270 in 2018, 65,206 in 2019, and 59,129 in 2020, amounting to 192,605 deaths over the three-year period. Among these deaths, 35%, 34%, and 31% occurred in 2018, 2019, and 2020, respectively. Children under one year of age accounted for 53% of the deaths. Males accounted for the majority of deaths (63%). When analyzing race/ethnicity data, the reported deaths indicated that individuals identified as mixed race accounted for 53% of fatalities, whereas those of unknown race/ethnicity comprised 5%. The lack of comprehensive race/ethnicity data in DATASUS is a limitation of the present study. Hospital deaths accounted for 74% of these deaths.

The total number of deaths among individuals diagnosed with CP and other paralytic syndromes across the five regions was 745 in 2018, 740 in 2019, and 539 in 2020, totaling 2024 over the three-year period. Of these, 37% occurred in 2018, 37% in 2019, and 27% in 2020. Adolescents aged 15–19 years accounted for 34% of the deaths; children aged 10–14 years for 28%, children aged 5–9 years for 17%, children aged 1–4 years for 16%, and infants under 1 year for 5%. Males accounted for the majority of deaths (58%). White individuals accounted for 55% of the deaths, while those of unknown race/ethnicity accounted for 3%. The lack of data on race/ethnicity in the DATASUS represents a limitation of this study. Hospital mortality accounted for 62% of the study records.

The analysis of health trend data in Brazil from 2018 to 2021 reveals regional patterns that highlight significant disparities in health access, particularly in regions with lower Human Development Index (HDI) scores. Despite the expansion of healthcare infrastructure, as evidenced by an increase in the number of facilities and medical professionals, this growth has not been sufficient to fully address the high rates of hospitalization and mortality in specific regions, particularly in the north and northeast. The analysis revealed the following: There was a notable expansion in the healthcare infrastructure, with a considerable increase in the number of healthcare facilities and medical practitioners across all regions, particularly in the south and southeast. Despite an increase of 261.77 in the number of establishments and 297.4 in the number of physicians, the northern region continues to experience elevated rates of hospitalization and mortality. It is imperative that not only more healthcare facilities be established but also that these facilities be fully equipped to provide specialized care. It is imperative that secondary and tertiary healthcare units be expanded in the north and northeast, with an emphasis on intensive care and diagnostic services. This is because the capacity to manage complex conditions such as cerebral palsy remains limited. Continuing training programs for medical professionals and other healthcare workers in these regions are also crucial for improving the quality of care.Significant regional disparities were identified in the treatment of complex conditions, with a notable increase in hospitalizations for cerebral palsy and paralysis in the north, while the northeast demonstrated a declining trend (VPA = 4.7, *p* < 0.001). This indicates that the northeast may have implemented effective early diagnosis and chronic condition management strategies, in contrast to the north, where demand continues to increase. One potential immediate solution would be to replicate the most effective management practices for complex conditions (such as cerebral palsy) in regions with a decreasing trend, including the northeast, the north, and other regions with increased hospitalizations. This may entail an expansion of access to rehabilitation and specialized support services at the primary and tertiary levels, as well as the establishment of national programs designed to equip multidisciplinary teams with the skills to address these conditions.Mortality rates continue to rise in the northern, northeastern, and central regions of the country, with a VPA of 97.95, 86.94, and 81.88, respectively. However, the south and southeast regions demonstrated a stationary trend, indicating that the growth in mortality has been contained. To reduce infant and general mortality rates in regions with the highest growth in deaths, it is imperative to strengthen primary healthcare, increasing the scope of preventive programs, particularly in rural and remote areas. It is also essential to reinforce vaccination campaigns, promote early diagnosis, and combat infectious diseases and chronic conditions. Furthermore, improvements in emergency and urgent care services, with a focus on optimizing patient flow and rapid hospital response times, may contribute to a reduction in mortality rates.The elevated hospitalization rates for all pathologies in the north (3951.83 VPA) and northeast (3687.98 VPA) regions indicate that these regions face significant challenges in accessing preventive and ambulatory care services. This may be indicative of a lack of robust primary care networks in these regions, leading to an increase in preventable hospitalizations. One potential solution is the reinforcement of primary healthcare networks in the north and northeast regions. This may be achieved by expanding family health programs and strengthening the role of community health agents, who may provide more accessible care to the population and prevent unnecessary hospitalizations. Moreover, investment in health information technology, such as electronic medical records and telehealth systems, has the potential to enhance chronic disease management and preventive monitoring, thereby reducing hospitalizations.Sustainable investment in health: Despite the positive growth in the number of medical professionals and facilities, the data analysis indicates that merely increasing the number of healthcare workers and infrastructure is insufficient. The regions that are most in need of health services continue to experience the highest rates of hospitalizations and mortality. To ensure a more profound and enduring impact, investments must be strategically planned, with an emphasis not only on quantity but also on the equitable distribution of health resources. This entails directing investments towards the construction of health centers in rural and peripheral areas, in addition to ensuring that existing hospitals and clinics are adequately equipped with advanced medical technology. The development of a local healthcare workforce, with incentives for the retention of medical practitioners and specialists in remote areas, may prove an effective approach to addressing regional disparities.

This study demonstrates that, despite advancements in the expansion of health infrastructure across all regions of Brazil, regional disparities continue to exert a detrimental impact on population health in the north, northeast, and center-west regions. The reinforcement of primary care, the implementation of superior chronic condition management practices, and the enhancement of emergency and urgent care services are pivotal steps for the reduction in hospitalizations and mortality in these regions. It is imperative that investment in infrastructure extend beyond mere numerical expansion to encompass considerations of service quality, continuous professional development, and the integration of novel health technologies. These tangible solutions, grounded in the analyzed data, can serve as a compass for future public policies and investments, with the aim of fostering a more equitable access to healthcare across the entire national territory.

## 4. Discussion

Significant disparities were evident when comparing the healthcare infrastructures available to populations across different regions of Brazil. Among the analyses described in this study, two notable examples emerged: the north had 3% more inhabitants than the center-west, yet the latter had 103% more healthcare facilities and 138% more physicians than the north. Similarly, the southeast had 27% more inhabitants than the north but boasted 163% more healthcare facilities and 228% more physicians. These discrepancies remained consistent throughout the four years analyzed in this study (2018–2021).

Social inequality refers to economic and social disparities among various groups within a society. In highly unequal societies, marginalized groups often have limited access to essential resources, including healthcare. This limited access can lead to challenges in obtaining vaccinations, hospital care, and medical treatments for children from low-income families, further exacerbating health inequalities [24]. It is crucial that statistical data accurately reflect the reality of each region to ensure that specialized services tailored to the needs of the population are provided in areas where children are born and raised. Other studies report underreporting and regional disparities in Brazil [25,26,27].

The analysis of health service utilization patterns is complex because of the interplay between service availability, accessibility, and sociodemographic and epidemiological characteristics of users. Changes in the population’s demographic and epidemiological profiles, transformations in healthcare organizations, and the ongoing incorporation and demand for new technologies require increasingly detailed studies of hospital care and the development of tools to support the management of both the SUS and Supplementary Healthcare System [28]. Comparing healthcare service utilization data before and after the onset of the COVID-19 pandemic is crucial for understanding changes in healthcare demands and service delivery patterns [3].

By correlating data from DATASUS with the Human Development Index (HDI) for each Brazilian region, it is possible to contribute to the formulation of public policies [28]. When correlating the HDI with DATASUS data, it becomes evident that regions with lower HDI scores also have a less developed healthcare support infrastructure [16]. The findings of this study on socioeconomic and healthcare indicators demonstrate an expansion in the availability of healthcare facilities and physicians in Brazil, particularly in 2020 (Appendix A), which coincides with the onset of the COVID-19 pandemic. However, the uneven distribution of healthcare services across Brazilian regions remained unchanged, supporting the results of Oliveira et al. [29] and Costa et al. [30].

Disparities in healthcare infrastructure affect hospitalization rates [31,32]. However, there was a clear decrease in hospitalizations starting in 2020. The COVID-19 pandemic appears to have affected pediatric epidemiology by reducing the number of medical visits and hospitalizations [33,34]. Mapping the data available in DATASUS by age group and region can influence the implementation of specialized services in areas with specific demands, such as medical and therapeutic treatments for CP. Because CP is typically diagnosed in childhood, limited access to healthcare may result in missed diagnoses, consequently hindering access to appropriate rehabilitation treatments.

When comparing the hospitalizations of individuals with CP and other paralytic syndromes in the northern region with those in other regions, it is evident that the northern region has significantly fewer hospitalizations. A study conducted by the Association for Assistance to Disabled Children (AACD, São Paulo unit) profiled the population served by the institution and examined the socioeconomic status of patients who participated in Binha et Bezerra’s study [6]. The analysis focused on territoriality, considering patients’ birthplaces and origins according to the IBGE’s standard geographical divisions. There was a predominance of families who reported relocating to São Paulo seeking improved access to specialized and multidisciplinary healthcare services. Regarding the socioeconomic profile, 55.5% of families did not receive any social benefits, 36.6% received some form of social assistance, and in 7.7% of cases, this information was unavailable. In terms of access to specialized medical care prior to initial screening/consultation at the AACD, 62.4% of cases had previously consulted a neurologist/neurosurgeon at least once, 50.3% had seen a pediatrician, 15.1% had visited an orthopedist, and 97.7% had never consulted a physiatrist or specialist in physical medicine or rehabilitation. This finding aligns with data from another study that associated children and adolescents with CP in northeastern Brazil and their families with individuals living in extreme poverty [35].

The analysis of the discrepancy between hospitalizations by place of residence and place of occurrence implies that individuals from the north may have sought medical care in other regions. This conclusion is not definitive, as it is impossible to pinpoint exactly where patients who migrated from other regions received treatment, given fluctuations in hospitalization data. This temporal sample maintained consistent regional disparities, suggesting its relevance to current conditions.

Considering the above discussion, it is essential to identify the strengths and weaknesses of this study. In terms of this study’s strengths, the use of an ecological time-series research design enabled an analysis of changes over time and comparisons across different regions. This approach provides a general understanding of health-inequality dynamics over different periods [36]. Based on extensive population data from the Ministry of Health, this study offers a comprehensive perspective and enables the analysis of national trends. It specifically addresses data on individuals with CP and other paralytic syndromes, allowing for a more in-depth analysis and understanding of health inequities in these cases. The findings of this study suggest significant regional differences in hospitalizations, which could serve as a powerful tool to initiate discussions, drive changes in health policies, and tailor care to address specific needs and inequities in different areas.

Health disparities across Brazilian regions are not an isolated phenomenon; they are closely linked to human and social development indicators. Therefore, studies such as this one update the landscape based on official data and underscore the importance of statistical information that accurately represents the reality of each region. This ensures that public and private healthcare policies effectively address the population’s needs.

In regard to the vulnerable points: This study is based on secondary data provided by the Brazilian Ministry of Health’s official health information system, DATASUS. Although this database is a valuable tool for epidemiological analysis, it is essential to acknowledge its limitations. Firstly, the utilization of secondary data entails that the quality and completeness of the information are contingent upon the accuracy with which it was recorded. In many instances, the data may be incomplete or inaccurate, which could potentially impact subsequent analyses and conclusions. A notable example is the significant absence of information regarding race/ethnicity, which constrains the capacity to fully explore the sociodemographic variables that could potentially influence the results. Furthermore, there is a possibility of the underreporting of hospitalizations and deaths, particularly in conditions with challenging diagnoses, such as cerebral palsy and other paralytic syndromes. This may result in an underestimation of the prevalence and impact of these diseases.

In light of the significance of conducting studies that demonstrate the disparities in access to healthcare in diverse regions of a developing country like Brazil, this study aims to contribute to an enhancement in that access, particularly in populations residing in areas with a lower Human Development Index (HDI). Furthermore, this study underscores the necessity for a continuous improvement in health data collection systems, such as DATASUS, to ensure that public policies are based on more precise and comprehensive data, thereby promoting equity in health access.

## 5. Conclusions

There are significant regional disparities in access to healthcare professionals and medical facilities in Brazil, leading to variations in hospitalization rates for children and adolescents. This is particularly pronounced in the northern region, where 50% of individuals with cerebral palsy and other paralytic syndromes seek treatment in other regions due to limited local resources. These disparities suggest that health policies should prioritize investments in healthcare infrastructure, particularly in underserved regions such as the north, to ensure more equitable access to care.

## Data Availability

https://datasus.saude.gov.br/ (accessed on 22 October 2022).

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
