# Peer review of "Impact of Health Access on Hospitalization of Children and Youth with Cerebral Palsy and Paralytic Syndromes in Brazilian Regions"

_ijerph, 2024, doi:10.3390/ijerph21101286_

Round 1

Reviewer 1 Report (Previous Reviewer 1)

Comments and Suggestions for Authors

The Authors following the reviewers comments and suggestions re-wrote with accuracy and appropriateness the paper that as to me doesn't need further modifications or adds.

Author Response

Dear Reviewer,Thank you very much for your attention and positive recommendation to publish the article written by me and Dr. Rubens. 
We are clear that this is a work that can contribute to public health policies in Brazil, especially benefiting socioeconomically disadvantaged people. 
Sincerely,                                                                                                                                                     

Silvia

Reviewer 2 Report (Previous Reviewer 2)

Comments and Suggestions for Authors

The peer-reviewed article ‘Impact of Health Access on Hospitalization of Children and Youth with Cerebral Palsy and Paralytic Syndromes in Brazilian Regions’ addresses the important topic of regional inequalities in access to health care in Brazil and their impact on hospitalizations and mortality among children with cerebral palsy (CP) and other paralytic syndromes. Although the topic is important and fits well with current discussions on social inequalities, the article has several important limitations that need to be improved before it can be accepted for publication.

Firstly, the study's methodology, although based on available secondary data, suffers from some shortcomings in precision and detail. The authors base their analysis on data from DATASUS, but there is no detailed discussion of potential sources of error, such as underestimation of hospitalisations or lack of complete demographic information. An example of this is the lack of data on race/ethnicity in a significant proportion of the cases analysed, which significantly limits the conclusions that can be drawn from the analysis.

Another limitation is the statistical analysis which, despite the use of regression models, does not provide sufficient information on the significance of correlations between variables. The article indicates significant differences between regions, but does not provide a sufficiently comprehensive analysis to understand the deeper causes of these differences. There is also a lack of information on potential confounding factors that could affect the results.

Furthermore, the conclusions presented are too general in places and not always supported by relevant data. The article points to the need for investment in health infrastructure in low HDI regions, but does not provide a concrete analysis of which elements of health infrastructure are most deficient and require immediate intervention.

I therefore recommend that the authors consider the following amendments: a more detailed discussion of the research methodology, with an emphasis on the potential limitations of the data used; a strengthening of the analytical section, especially in terms of statistics, to better explain the correlations presented; and more detailed conclusions that directly address the results obtained and propose concrete solutions to improve access to healthcare in Brazil.

In its current form, publication of the article is not recommended. It requires significant revisions to make it a valuable contribution to the scientific literature and practical efforts to improve public health in Brazil. Only after the suggested changes have been made will it be possible to consider its resubmission for evaluation.

Author Response

Dear Reviewer,

thank you very much for your valuable contributions to the article written by myself and Dr. Rubens. We have carefully analyzed all the points you made and tried to respond to them in detail, going even further into the content of this work. Our main objective is to promote improvements in access to healthcare in Brazil, with special attention to the most socio-economically vulnerable situations. Your suggestions were fundamental to improving our approach and ensuring that the study can generate a significant impact in the field.

Thank you again for your careful review and for the opportunity to improve the work

Sincerely,

Silvia e Dr. Rubens

Comments 1. A more detailed discussion of the research methodology, with an emphasis on the potential limitations of the data used (the study's methodology, although based on available secondary data, suffers from some shortcomings in precision and detail. The authors base their analysis on data from DATASUS, but there is no detailed discussion of potential sources of error, such as underestimation of hospitalizations or lack of complete demographic information. An example of this is the lack of data on race/ethnicity in a significant proportion of the cases analyzed, which significantly limits the conclusions that can be drawn from the analysis).

Response 1: In regard to the vulnerable points: This study is based on secondary data provided by the Brazilian Ministry of Health's official health information system, DATASUS. Although this database is a valuable tool for epidemiological analysis, it is essential to acknowledge its limitations. Firstly, the utilization of secondary data entails that the quality and completeness of the information are contingent upon the accuracy with which it was recorded. In many instances, the data may be incomplete or inaccurate, which could potentially impact subsequent analyses and conclusions. A notable example is the significant absence of information regarding race/ethnicity, which constrains the capacity to fully explore the sociodemographic variables that could potentially influence the results. Furthermore, there is a possibility of underreporting of hospitalizations and deaths, particularly in conditions with challenging diagnoses, such as cerebral palsy and other paralytic syndromes. This may result in an underestimation of the prevalence and impact of these diseases.

In light of the significance of conducting studies that demonstrate the disparities in access to health care in diverse regions of a developing country like Brazil, this study aims to contribute to the enhancement of that access, particularly in populations residing in areas with a lower Human Development Index (HDI). Furthermore, this study underscores the necessity for continuous improvement of the health data collection system, such as DATASUS, to ensure that public policies are based on more precise and comprehensive data, thereby promoting equity in health access.

Comments 2. A strengthening of the analytical section, especially in terms of statistics, to better explain the correlations presented (another limitation is the statistical analysis which, despite the use of regression models, does not provide sufficient information on the significance of correlations between variables. The article indicates significant differences between regions, but does not provide a sufficiently comprehensive analysis to understand the deeper causes of these differences. There is also a lack of information on potential confounding factors that could affect the results).

Response 2: Table 7 illustrates the evolution of health indicators in Brazil between 2018 and 2021. It presents the annual percentage change (APC) of key metrics, including the number of establishments, healthcare professionals, hospitalizations, and deaths, disaggregated by region.

A further analysis is presented according to category.

  1. Healthcare Establishments:

All regions exhibited an upward trajectory in the number of healthcare establishments, with a notable positive acceleration (p < 0.001 for the majority of regions).

The South exhibited the highest APV (935.36), while the North demonstrated the lowest growth (261.77), both with an upward trend.

The Durbin-Watson test indicates that the values are within an acceptable range, suggesting that there is no significant autocorrelation in the model residuals, with the exception of the Southeast, which exhibited a value of 2.95, indicating the potential for serial correlation.

  1. Number of Doctors:

Furthermore, all regions demonstrated a notable increase in the number of medical practitioners, with an APV ranging from 297.4 (in the North) to 955.43 (in the Southeast).

As with establishments, the Southeast and South have the highest growth rates in the number of doctors.

All Durbin-Watson values indicate an upward trend.

  1. Hospitalizations for All Conditions (Hosp_ALL_place):

All regions exhibited a notable increase in hospitalizations for all pathologies, with the North region demonstrating the most pronounced upward trend, as indicated by the highest APV (3951.83), and the South exhibiting the least pronounced trend (3990.69).

The Durbin-Watson value for all regions indicates the absence of serial correlation, with a consistent upward trend.

  1. Hospitalizations for Cerebral Palsy (Hosp_CP_place):

The data indicate an upward trend in hospitalizations for cerebral palsy in nearly all regions, with the exception of the Northeast, where a significant downward trend was observed (APV = 4.7; p < 0.001).

The Midwest region exhibited the most pronounced percentage variation (APV = 10).

The Durbin-Watson test indicates the potential for serial correlation in the Northeast region (value of 2.66), which merits further investigation.

  1. Mortality from all causes (Dea_ALL_place):

Furthermore, a similar upward trend was observed in deaths from all pathologies in the North, Northeast, and Central-West regions, with the highest APV in the North (97.95).

In the Southeast and South, the trend is stable, with Durbin-Watson values also indicating the absence of autocorrelation in these cases.

It is noteworthy that, although the APV for these regions is lower, the values are nevertheless marginally significant, suggesting a stabilization in deaths.

The data indicate a trend towards an increase in both health infrastructure and the number of doctors in all regions of Brazil, which reflects ongoing investments in the health sector. However, the increases in hospitalizations and deaths, especially in the North and Northeast regions, suggest that these structural advances have not yet been sufficient to reverse health conditions in these areas, particularly in the context of chronic diseases such as cerebral palsy.

The stabilization of mortality rates in the Southeast and South may indicate a reduction in risk factor control or an improvement in the management of chronic diseases such as cerebral palsy. However, the continued growth in hospitalizations for cerebral palsy in the Midwest requires further investigation, as it may reflect improvements in diagnosis or an increase in incidence.

Comments 3. More detailed conclusions that directly address the results obtained and propose concrete solutions to improve access to healthcare in Brazil (furthermore, the conclusions presented are too general in places and not always supported by relevant data. The article points to the need for investment in health infrastructure in low HDI regions but does not provide a concrete analysis of which elements of health infrastructure are most deficient and require immediate intervention.).

Response 3. The analysis of health trend data in Brazil from 2018 to 2021 reveals regional patterns that highlight significant disparities in health access, particularly in regions with lower Human Development Index (HDI) scores. Despite the expansion of healthcare infrastructure, as evidenced by an increase in the number of facilities and medical professionals, this growth has not been sufficient to fully address the high rates of hospitalization and mortality in specific regions, particularly in the North and Northeast.

  1. There was a notable expansion in the healthcare infrastructure, with a considerable increase in the number of healthcare facilities and medical practitioners across all regions, particularly in the South and Southeast. Despite an increase of 261.77 in the number of establishments and 297.4 in the number of physicians, the Northern region continues to experience elevated rates of hospitalization and mortality. It is imperative that not only more health care facilities be established, but also that these facilities be fully equipped to provide specialized care. It is imperative that secondary and tertiary health care units be expanded in the North and Northeast, with an emphasis on intensive care and diagnostic services. This is because the capacity to manage complex conditions such as cerebral palsy remains limited. Continuing training programs for medical professionals and other healthcare workers in these regions are also crucial for improving the quality of care.
  2. Significant regional disparities were identified in the treatment of complex conditions, with a notable increase in hospitalizations for cerebral palsy and paralysis in the North, while the Northeast demonstrated a declining trend (VPA = 4.7, p < 0.001). This indicates that the Northeast may have implemented effective early diagnosis and chronic condition management strategies, in contrast to the North, where demand continues to in-crease. One potential immediate solution would be to replicate the most effective management practices for complex conditions (such as cerebral palsy) in regions with a decreasing trend, including the Northeast, the North, and other regions with increased hospitalizations. This may entail an expansion of access to rehabilitation and specialized support services at the primary and tertiary levels, as well as the establishment of national pro-grammes designed to equip multidisciplinary teams with the skills to address these conditions.
  3. Mortality rates continue to rise in the northern, northeastern, and central regions of the country, with a VPA of 97.95, 86.94, and 81.88, respectively. However, the South and Southeast regions demonstrated a stationary trend, indicating that the growth in mortality has been contained. To reduce infant and general mortality rates in regions with the highest growth in deaths, it is imperative to strengthen primary health care, in-creasing the scope of preventive programs, particularly in rural and remote areas. It is also essential to reinforce vaccination campaigns, promote early diagnosis, and combat infectious diseases and chronic conditions. Furthermore, improvements in emergency and urgent care services, with a focus on optimizing patient flow and rapid hospital response times, may contribute to a reduction in mortality rates.
  4. The elevated hospitalization rates for all pathologies in the North (3951.83 VPA) and Northeast (3687.98 VPA) regions indicate that these regions face significant challenges in accessing preventive and ambulatory care services. This may be indicative of a lack of robust primary care networks in these regions, leading to an increase in preventable hospitalizations. One potential solution is the reinforcement of primary health care networks in the North and Northeast regions. This may be achieved by expanding family health pro-grams and strengthening the role of community health agents, who may provide more accessible care to the population and prevent unnecessary hospitalizations. Moreover, in-vestment in health information technology, such as electronic medical records and tele-health systems, has the potential to enhance chronic disease management and preventive monitoring, thereby reducing hospitalizations.
  5. Sustainable Investment in Health: Despite the positive growth in the number of medical professionals and facilities, the data analysis indicates that merely increasing the number of healthcare workers and infrastructure is insufficient. The regions that are most in need of health services continue to experience the highest rates of hospitalizations and mortality. To ensure a more profound and enduring impact, investments must be strategically planned, with an emphasis not only on quantity but also on the equitable distribution of health resources. This entails directing investments towards the construction of health centers in rural and peripheral areas, in addition to ensuring that existing hospitals and clinics are adequately equipped with advanced medical technology. The development of a local healthcare workforce, with incentives for the retention of medical practitioners and specialists in remote areas, may prove an effective approach to addressing regional disparities.

The study demonstrates that, despite advancements in the expansion of health infrastructure across all regions of Brazil, regional disparities continue to exert a detrimental impact on population health in the North, Northeast, and Center-West regions. The reinforcement of primary care, the implementation of superior chronic condition management practices, and the enhancement of emergency and urgent care services are pivotal steps for the re-duction of hospitalizations and mortality in these regions. It is imperative that investment in infrastructure extend beyond mere numerical expansion to encompass considerations of service quality, continuous professional development, and the integration of novel health technologies. These tangible solutions, grounded in the analyzed data, can serve as a compass for future public policies and investments, with the aim of fostering a more equitable access to healthcare across the entire national territory.

This manuscript is a resubmission of an earlier submission. The following is a list of the peer review reports and author responses from that submission.

Round 1

Reviewer 1 Report

Comments and Suggestions for Authors

The Authors present a very detailed paper: " Impact of access to healthcare on Pediatric hospitalization in Brazilian Regions" not easy to be followed.

Introduction as well as Objectives are clear and well done.

Materials and Methods: Study site is good, Study Population is good, Data Stratification is too long and difficult to understand. Please remove from the text and move as appendix

Results: I suggest to move the complete table as appendix and in the text insert only one table reporting only the significant results (with a statistical value).

Discussion: it is too much dedicated to the Brazilian situation. I understand that the study reports the Brazilian experience but what is the take home message for readers that should be adopted in a practical way?

Author Response

Point 1: The Authors present a very detailed paper: " Impact of access to healthcare on Pediatric hospitalization in Brazilian Regions" not easy to be followed.

Response 1: Dear Reviewer, I greatly appreciate your contributions to my work; they have been instrumental in improving the manuscript.

I have revised the text to make it more fluent in reading. The modifications are highlighted in blue within the manuscript.

Introduction as well as Objectives are clear and well done. Grateful

Point 2: Materials and Methods: Study site is good, Study Population is good, Data Stratification is too long and difficult to understand. Please remove from the text and move as apêndix.

Response 2: I have modified the data stratification, summarizing the process. I will include the complete process in the appendix.

Point 3: Results: I suggest to move the complete table as appendix and in the text insert only one table reporting only the significant results (with a statistical value).

Response 3: I removed all tables from the text (placed them in an appendix) and measured only the most relevant data in the manuscript.

Point 4: Discussion: it is too much dedicated to the Brazilian situation. I understand that the study reports the Brazilian experience but what is the take home message for readers that should be adopted in a practical way?

Response 4:

This publication aimed to contribute to the understanding of health inequalities in the pediatric population by analyzing disparities in healthcare for children in Brazil, with a particular focus on the context of neurological diseases. Specifically, the research focused on Cerebral Palsy and other paralytic syndromes, investigating their relationship with access to healthcare services in different regions of the country. The results of this study have significant implications for health policy formulation and healthcare planning, allowing for the identification of areas of inequality and specific needs that should be addressed to improve the quality of life and health outcomes of children affected by these conditions.

The results obtained are essential indicators for monitoring how official health structure records impact the treatment of individuals within the age range described in this study, particularly in hospitalizations of those with Cerebral Palsy and other paralytic syndromes. It is crucial that statistical data accurately reflect each region's reality to ensure that public and private health policies effectively meet the population's needs.

Reviewer 2 Report

Comments and Suggestions for Authors

Dear Authors,

I present the publication submitted to me for review on the relationship between health structure inequalities and hospitalizations and deaths among the pediatric population in different regions.

The publication presented here aimed to investigate the relationship between health structure inequalities and the number of hospitalizations and deaths among the pediatric population in different regions. The authors applied an ecological temporal design using population-based secondary data they obtained from DATASUS, the Brazilian Ministry of Health. The data included the population of children between the ages of zero and 19, health facilities, physicians working in those facilities, hospitalizations and deaths related to diseases listed in the International Classification of Diseases. There was a particular focus on those diagnosed with cerebral palsy, including cerebral palsy, and other sclerosis syndromes. Data was collected from January 2018 to December 2021, using TABNET for data collection and Excel for tabulation and analysis.

The results of the analysis of health structure data in different regions showed significant disparities, especially in the North and Northeast regions compared to other areas. These disparities prompted the decision to conduct a comparative assessment between these regions and the rest of the country. The findings of the study indicate that a significant difference of 50% in the number of hospitalizations by place of residence was observed in the Northern region compared to place of care for people with cerebral palsy and other sclerosis syndromes. This result indicates the need for relocation to access appropriate medical treatment.

From the mini-comments, I ask that the tables be adjusted to meet editorial requirements and that the literature be transcribed according to the appropriate scheme. A more meticulous demonstration of the study's strengths and weaknesses is also necessary. In conclusion, this publication adds value by analyzing health care disparities among children in Brazil. However, like any study, some limitations must be considered when interpreting the results and conclusions.

Strengths of the study:

1. **The use of ecological temporal design:** The use of this type of research design makes it possible to analyze changes over time and compare different regions. This approach can provide a general understanding of the dynamics of health care inequality over different periods.

2 **Use of population data:** The study relies on a large amount of population data obtained from the Brazilian Ministry of Health. This provides a broader perspective and enables analysis of trends at the country-wide level.

3. **Disease-specific focus:** The study focuses on people with cerebral palsy and other sclerosis syndromes. This allows for a more thorough analysis and understanding of health care inequities for these specific cases.

4 **Indicates the need to restructure health policies:** Findings from the study suggest that there are significant differences in hospitalizations by region. This could be a powerful tool for initiating discussions and changes in health policies to tailor care to specific needs and inequities in different areas.

Weaknesses of the study:

 1. **Risk of underestimation of hospitalizations and deaths:** The authors highlight the hypothesis of underestimation of hospitalizations and deaths due to problems in the diagnostic process for diseases such as cerebral palsy. This could affect the accuracy and completeness of the data and the interpretation of the results.

2. **Methodological Limitations:** Lack of information on specific diagnostic methods and treatments and other variables that may affect the results may limit the overall strength of the study's conclusions.

3. **Lack of control over the quality of source data:** The study relies on existing data from a secondary source. There is no assurance that these data are complete and accurate, which may affect the results and conclusions.

4. **Limited generalizability** The results apply to a specific country and its various regions. The results may be difficult to generalize to other countries with different health systems and socioeconomic conditions.

5. ** Lack of analysis of the causes of inequality:** The study focuses on identifying inequalities, but does not analyze the deeper causes of these differences, which could provide a more comprehensive understanding of the problem.

This publication makes a valuable contribution to understanding health inequalities among the pediatric population, particularly in the context of neurological diseases. Focusing on the specific case of cerebral palsy and its relationship to access to medical care in different regions provides important lessons for health policy and care planning. It is worth noting, however, that further research may be needed to better understand the causes of these disparities and to develop effective strategies for reducing disparities in pediatric health care.

Author Response

Dear Reviewer, I greatly appreciate your contributions to my work; your dedication was crucial for enhancing the manuscript.

Point 1: From the mini-comments, I ask that the tables be adjusted to meet editorial requirements and that the literature be transcribed according to the appropriate scheme.

Response 1: I removed all tables from the text (placed them in an appendix) and included in the manuscript only the most relevant data.

Point 2: A more meticulous demonstration of the study's strengths and weaknesses is also necessary.

Response 2: I appreciate the highlighting of the strengths and weaknesses of the work. Regarding the weaknesses, please strive to correct the feasible aspects as described below.

In conclusion, this publication adds value by analyzing health care disparities among children in Brazil. However, like any study, some limitations must be considered when interpreting the results and conclusions.

Strengths of the study:     

Point 3: 1. **Risk of underestimation of hospitalizations and deaths:** The authors highlight the hypothesis of underestimation of hospitalizations and deaths due to problems in the diagnostic process for diseases such as cerebral palsy. This could affect the accuracy and completeness of the data and the interpretation of the results.

Response 3: Unfortunately, this is a weakness of the study. There are discrepancies between the numbers of doctors reported by DATASUS and IBGE, both government sources.

Point 4:* 2. **Methodological Limitations:** Lack of information on specific diagnostic methods and treatments and other variables that may affect the results may limit the overall strength of the study's conclusions.

Response 4: Unfortunately, we do not have specific data about the number of people with Cerebral Palsy and other paralytic syndromes. Since this is a condition where symptoms manifest in childhood, the lack of healthcare infrastructure can impact the accurate diagnosis of children.

Point 5: 3. **Lack of control over the quality of source data:** The study relies on existing data from a secondary source. There is no assurance that these data are complete and accurate, which may affect the results and conclusions.

Response 5: Yes, it's important, including the comparison with other officially published data, as in Table 5 attached, since official health data impact the implementation of public policies.

Point 6: 4. **Limited generalizability** The results apply to a specific country and its various regions. The results may be difficult to generalize to other countries with different health systems and socioeconomic conditions.  
Response 6: Yes, but they make all the difference in how social inequality impacts the health of millions of children.

Point 7: 5. ** Lack of analysis of the causes of inequality:** The study focuses on identifying inequalities, but does not analyze the deeper causes of these differences, which could provide a more comprehensive understanding of the problem.
Response 7: I delved deeper into research in this direction, seeking articles that extensively analyzed the causes of social inequalities.

Reviewer 3 Report

Comments and Suggestions for Authors

Thank you for the opportunity to review your manuscript. 

Overall this is a useful paper to inform health infrastructure considerations at a country level. The manuscript itself is long with many redundant discourse that does not add value to the paper. The tables, charts and figures need to be given consideration and formatted for consistency throughout the manuscript, but also across the journal (with other articles). 

You use CP and cerebral palsy interchangeably, please be consistent.

Title: Since the focus is on patients with CP and other paralytic syndromes, please include this in the title. 

Abstract: the objective needs to include the population you are interested in. The results mention modification based on the results. All methods for the entire study should be in the methods even though the temporal relationship does not fit. The conclusion should not introduce new material, please put results (eg 50% difference) in the results part. 

Introduction: 

Objectives: the wording is unclear for general objectives. 

Methods

In study population you say 2 months before the COVID-19 pandemic...i think you mean 2 years? 

Your data collection descriptions are better placed in supplementary material. 

Results

Table 1. should be formatted so the title is on the same line, 2018^a doesnt have a reference

Is figure 1. necessary?

Table 3. Do not circle figures in tables

Be careful with introducing discussion in your results eg "This analysis suggests that individuals from the North may have migrated to other regions in search of medical care. The conclusion cannot be precise as it is impossible to determine with certainty in which region patients who migrated from other regions were treated due to the variation in admission numbers."

I didnt see any statistical analysis of the findings - i would assume that the disparity between place of residence and place of occurence to be significant, but it would be good for this to be objectively proven with some statistical analysis to inform. 

Discussion

Your last 2 paragraphs discuss limitations - you should start the paragraphs by saying that there are some limitations to the study including...

Conclusion

Good

Comments on the Quality of English Language

The language is largely fine, grammar needs to be corrected in some areas and there are some spelling mistakes throughout

Author Response

Dear Reviewer, I greatly appreciate your contributions to my work; your dedication was instrumental in improving the manuscript.

Point 1: Overall this is a useful paper to inform health infrastructure considerations at a country level. The manuscript itself is long with many redundant discourse that does not add value to the paper.

Response 1: I rewrote the methods, results, discussion, and conclusion to make the text flow more smoothly in reading. The modifications are highlighted in blue in the manuscript.

Point 2: The tables, charts and figures need to be given consideration and formatted for consistency throughout the manuscript, but also across the journal (with other articles).

Response 2: I removed all tables from the text (placed them in the appendix) and only included the most relevant data in the manuscript.

Point 3: You use CP and cerebral palsy interchangeably, please be consistent.

Response 3: I corrected the terminology.

Point 4: Title: Since the focus is on patients with CP and other paralytic syndromes, please include this in the title.

Response 4: Impact of access to health on the hospitalization of children with Cerebral Palsy in Brazilian regions.

Point 5: Abstract: the objective needs to include the population you are interested in. The results mention modification based on the results. All methods for the entire study should be in the methods even though the temporal relationship does not fit. The conclusion should not introduce new material, please put results (eg 50% difference) in the results part.

Response 5: I made all the modifications to the article as per the instructions.

Point 6: Objectives: the wording is unclear for general objectives.

Response 6: I made the modification to the overall objectives.

Point 7: Methods - In study population you say 2 months before the COVID-19 pandemic...i think you mean 2 years?

Response 7: Yes, I corrected it to two years.

Point 8: Methods - Your data collection descriptions are better placed in supplementary material.

Response 8: I modified the data stratification, summarizing the process. I will include the complete process as an appendix.

Point 9: Results - Table 1. should be formatted so the title is on the same line, 2018^a doesnt have a reference.  Is figure 1. necessary? Table 3. Do not circle figures in tables.

Response 9: I removed all tables from the text (placed them in the appendix) and included in the manuscript only the most relevant data.

Point 10: Be careful with introducing discussion in your results eg "This analysis suggests that individuals from the North may have migrated to other regions in search of medical care. The conclusion cannot be precise as it is impossible to determine with certainty in which region patients who migrated from other regions were treated due to the variation in admission numbers."

Response 10: I revised the manuscript, moving this part of the text to the discussion section.

Point 11: I didnt see any statistical analysis of the findings - i would assume that the disparity between place of residence and place of occurence to be significant, but it would be good for this to be objectively proven with some statistical analysis to inform.

Response 11: The statistical service of the University Center at Faculdade de Medicina ABC assisted me during the statistical analysis and guided that a descriptive analysis be conducted, aiming to report and describe the events and variables of interest. The analysis of the rate of healthcare establishments and healthcare professionals per region was calculated annually from 2018 to 2021 as follows: the number of establishments (for each region) divided by the population aged between 0 and 19 years (for each region), multiplied by 100.000. This calculation was also used to measure doctors, hospitalizations, and deaths per region and for each year. This approach maintained the respective population proportions, in order to minimize the influence of external variations over the assessed time period.

Point 12: Discussion - Your last 2 paragraphs discuss limitations - you should start the paragraphs by saying that there are some limitations to the study including.

Response 12: I revised the discussion, incorporating the strengths and limitations of the study.